# A Qualitative Study on Prehabilitation before Total Hip and Knee Arthroplasties: Integration of Patients' and Clinicians' Perspectives

Somayyeh Mohammadi [1,2], Holly Reid [1,3], Wendy Watson [4], Morag Crocker [4], Julie M. Robillard [5,6], Marie Westby [4,7] and William C. Miller [1,3,*]

1   Department of Occupational Science and Occupational Therapy, Faculty of Medicine, University of British Columbia, Vancouver, BC V6T 1Z4, Canada; smh.mohammadi@gmail.com (S.M.); hollyreid@live.ca (H.R.)
2   Department of Psychology, Kingston University, London KT1 2EE, UK
3   Rehabilitation Research Program, GF Strong Rehabilitation Centre, Vancouver, BC V5Z 2G9, Canada
4   Vancouver Coastal Health, Vancouver, BC V5Z 1A1, Canada; wendy.watson@vch.ca (W.W.); morag.crocker@vch.ca (M.C.); marie.westby@vch.ca (M.W.)
5   BC Children's and Women's Hospital, Vancouver,BC V6H 3N1, Canada; Julie.robillard@ubc.ca
6   Division of Neurology, Department of Medicine, University of British Columbia, Vancouver, BC V6T 1Z4, Canada
7   Centre for Hip Health and Mobility, University of British Columbia, Vancouver, BC V5Z 1M9, Canada
*   Correspondence: bill.miller@ubc.ca

**Abstract:** To explore and integrate the perspectives of patients with hip and knee osteoarthritis (OA), their caregivers, and clinicians who are working with these patients about current preoperative rehabilitation ("prehab") content and delivery. Participants were individuals with hip ($n$ = 46) or knee OA ($n$ = 14), their family caregivers ($n$ = 16), and clinicians working with patients with hip/knee OA ($n$ = 15). In semi-structured interviews and focus groups, participants answered questions regarding barriers to accessing prehab, gaps in prehab content, learning preferences, and delivery formats. Interviews were audiotaped and transcribed verbatim. Data were analyzed using Qualitative Description method. Four main themes were identified: (1) "I didn't get any of that" discusses barriers in accessing prehab; (2) "I never got a definitive answer" highlights necessary information in prehab; (3) "better idea of what's going to happen" emphasizes the positive and negative aspects of prehab; (4) "a lot of people are shifting to online" describes participants' perspectives on online education. Our findings confirm the need for prehab education and the potential of online prehab education. The results inform the development of prehab educational modules based on users' input.

**Keywords:** osteoarthritis; total hip replacement; total knee replacement; prehab; surgery; preoperative education

## 1. Introduction

Patients can experience high levels of anxiety before and after major elective surgeries [1]. The preoperative anxiety can negatively impact patients' physical and psychological wellbeing and ultimately increase health care costs [2]. One of the main strategies to decrease preoperative anxiety is through preoperative rehabilitation ("prehab"). The main aim of prehab is to alleviate the anxiety before surgery and increase feelings of control in patients by prepare patients for surgery, surgical procedures, potential complications, and pre- and postoperative care and rehabilitation [3–5]. While various forms of prehab exist, the most common form of prehab involves exercise training and education [3,4].

Osteoarthritis (OA) is the most common joint disorder and one of the leading causes of pain and disability in adults [6]. Hip and knee OA affects up to 27% and 31.2% of the population, respectively [7,8]. When patients do not benefit from conservative treatments, total hip/knee arthroplasty (THA/TKA) is recommended [9]. The evidence showed that prehab for patients with hip and knee OA can reduce the length of stay and hospitalization

costs [10], minimize rehabilitation time [11], improve walking and functional capacity post-surgery [12], and reduce patient discomfort and preoperative anxiety [13–15]. However, the evidence for effectiveness of prehab in patients with hip and knee OA is mixed because prehab programs do not have the same content or topics [16–18], and therefore, currently, no universally adopted guideline is available that addresses the content of education that should be provided for patients before their total hip and knee arthroplasty [4].

Patient input regarding prehab content is vital for ensuring an effective and usable prehab. In addition to patients, family caregivers are also asked to provide the care for patients before and immediately after patients' surgery. Therefore, family caregivers can also be considered as the users of prehab education and therefore their perspective on the prehab content should be taken into account. Finally, it is also important to consider the perspective of clinicians who are interacting with patients undergoing hip and knee arthroplasties on prehab content and online delivery format, as they can contribute valuable knowledge regarding current gaps in the clinical setting [19,20]. However, in most cases, prehab is only based on clinicians' perspective and not patients and their family caregivers.

Even when prehab education is available, how it is provided differs substantially from one program to another. For example, clinicians (usually occupational therapist, physical therapists, and nurses) may invite various groups of patients (e.g., patients with hip OA and patients with knee OA) to the same session and provide the same information to both group because of lack of time, while patients with hip and knee OA should receive different prehab content. In addition, prehab can be delivered via in-person sessions, written material, or educational videos [16–18] which can impact the quality of the provided education. One platform that has recently, rapidly dominated the patient education field is online education or eHealth. eHealth is a promising way to address two substantial barriers in most prehab programs as it can offer interactive learning (unlike printed materials) and enable learners to engage over sustained periods (unlike attending few in-person sessions). However, the usability of eHealth for prehab programs needs further investigation.

The purpose of this study is to (1) explore the perspective of patients with hip or knee OA, the family caregivers of patients, and clinicians working with patients with hip or knee OA regarding prehab content, (2) investigate the barriers and facilitators affecting access to prehab, and (3) understand both patients' and clinicians' perspective on the online delivery format for prehab.

## 2. Materials and Methods

This qualitative study is based on focus groups and semi-structured interviews with three samples. Sample one included patients with hip OA and their family caregivers (not all patients had a family caregiver with them), sample two included patients with knee OA, and in sample three clinicians who were working with patients with hip and knee OA were recruited. To recruit the participants, advertisements and pamphlets were placed in several hospitals in the province of British Columbia (BC) in Canada. Participants were also recruited through email invitations that were sent by the Vancouver Coastal Health Research Institutes. In addition, OsteoArthritis Service Integration System (OASIS), which provides in-person education for patients undergoing joint arthroplasty in some parts of BC, added the advertisements to their newsletter. Finally, newspaper ads and twitter posts were also used to advertise for this study. This study has been approved by the Research Ethics Board of the University of British Columbia.

### 2.1. Eligibility Criteria

In order to be included in the study, patients were required to be 45 years of age or older with OA of the hip or knee, who were waiting for or had already undergone a THA or a TKA and were comfortable in speaking and reading in English. Patients with inflammatory arthritis were excluded from this study due to the differences between OA and inflammatory arthritis (e.g., chronic autoimmune disease, affected patients as young as 20, usually affects more than one joints). In addition, patients with THA were asked to

have a family caregiver attend with them if possible (no inclusion/exclusion criteria were in place for family caregivers).

Clinicians were eligible to participate if they had at least five years of clinical experience in working with patients with hip or knee OA before or after their total hip or knee arthroplasty, currently worked with patients with OA of the hip or knee before or after their THA or TKA, worked in BC, and were comfortable speaking and reading in English.

### 2.2. Data Collection

Before recruiting participants, the research team pilot-tested the interview guides (one focus group with eight participants for the THA sample, two semi-structured interviews for the TKA sample, and two semi-structured interviews for the clinician sample; Table S1 shows one of the interview guides used in this study). The data from these pilot interviews were used to refine the interview guides. The pilot interviews were not used in the analyses. All the interviews were audio recorded and transcribed verbatim. The accuracy of the transcripts was only reviewed by the research team and not by the participants. Field notes were made during each interview or focus group by the researchers. The data collection for each group of participants ended when the researchers agreed that the data reached a saturation point and no more new codes were developed [21].

Participants with hip OA and their family caregivers were assigned to a focus group which ranged in number of 2–9 participants, with nine focus groups total. Focus groups were conducted in both urban and rural areas in BC. There were five phone interviews performed for participants with OA who were unable to participate in the in-person focus groups due to mobility or other constraints (i.e., time restrictions). Participants with knee OA and clinicians participated in semi-structured interviews. The semi-structured interviews were either in-person or over the phone based on their availability and geographical location. To facilitate the collection of data and reduce the potential expenses and increase the feasibility of the data collection only semi-structured interviews were conducted with participants with knee OA and clinicians and family caregivers of participants with knee OA were not recruited.

All participants were asked questions about the available prehab for patients undergoing THA and TKA, the gaps in the current prehab, the benefits of prehab, and whether prehab could be done online. Each interview (45–60 min) and focus group (90 min) consisted of two main parts: the first, being preferred content and the learning preferences of patients with hip or knee OA in respect to presurgical education, and the second, focusing around the patients' preferred format for the presurgical education (e.g., images, text, different types of videos, and different types of questions) and feedback on an eHealth tool used to deliver prehab (more details about the second part have been provided in [22]. The data presented in this study are only based on the first part of the focus groups and semi-structured interviews.

S.M., W.W., M.C., and two research assistants conducted the focus groups and interviews for THA group. Authors S.M., H.R., and a student (P.H.) conducted the semi-structured interviews for THK group. Author 1 conducted the clinician interviews.

### 2.3. Research Team and Reflexivity

S.M. is a female with a PhD degree in health psychology and three years of experience in conducting research on patients with hip or knee osteoarthritis and four years of experience in conducting qualitative research. H.R. is a female and has a degree in Kinesiology and Master of Occupational Therapy and is working as an occupational therapist. H.R. has two years of experience in conducting qualitative research. W.W. is a female and a physical therapist with a bachelor's degree. She teaches the OsteoArthritis Service Integration System (OASIS) prehab education sessions for patients undergoing THA and TKA. M.C. is a female and an occupational therapist with a bachelor's degree. She also teaches the OASIS prehab education sessions for patients undergoing THA and TKA. J.M.R. is a female with a PhD degree in neuroscience. She has more than 10 years of experience

in working with patients with chronic illnesses. M.W. is a female with a PhD degree in physiotherapy who works with patients with joint replacement surgery and conducted qualitative and quantitative research during her career. W.C.M. is a male with a PhD in occupational therapy. He has more than 20 years of experience in conducting qualitative and quantitative research on patients with chronic illnesses including patients with hip and knee OA.

Before conducting the focus groups and interviews, S.M. or a research assistant in our center contacted the participants to check their eligibility and provided some information about the study and schedule an appointment for the participants. Team members and authors had no personal relationship with any of the participants. Participants only were informed about the interviewers' education, occupation, and reason for involvement in the research topic before the interviews. No other personal information was disclosed about the interviewers to the participants.

### 2.4. Data Analyses

The goal of this study was to describe the demands and gaps in prehab education and therefore Qualitative Description (QD [23]) was the selected methodology for analyzing the data in this study as it provides clear and comprehensive information for the researchers and clinicians on how to improve current practice and applies "low inferences interpretations" in comparison to other methodologies such as thematic analyses [24]. QD aims to help the researchers to provide a detailed description of the experience from the viewpoint of the participants, without applying unnecessary interpretations to participants' quotes [23,25].

Transcriptions of all focus groups and interviews were imported into a qualitative data analysis software, NVivo 12 (QSR International Pty Ltd, released in 2018) [26]. J.M.R., M.W., and W.C.M. trained the coders before proceeding with coding. S.M., W.W. and M.C. coded the focus groups and interviews of participants with hip OA and their family caregivers. HR and a master's student coded the interviews of participants with knee OA. S.M. and a research assistant coded the interviews with clinicians. To capture the perspective of each group of participants, three separate codebooks were developed.

For all the focus groups and interviews, first, each transcript was separately read by the coders to increase familiarity (two main coders were assigned to each interview). Then, each coder analyzed the first interview. Codes were compared after coders finished the analyzing of each interview. This approach was used for the first four interviews and to create the first draft of the codebooks. After developing the codebooks, the coders would send the codebook to the team members to receive their comments. The codebook was updated based on the comments received from the team members and then was used to code the rest of the interviews. Each coder then coded the next three interviews separately and the codes of these three were compared after both coders finished coding them. Finally, the remaining interviews were divided between two coders. Then, each coder reviewed the interviews coded by the other coder. In all the steps, the coders discussed any disagreement in person until they reached a consensus. If disagreement continued, a third coder was involved in reviewing and helping with the coding. The codebooks would be updated to reflect the newly developed codes. After a new version of the codebook was developed, the team members would be invited to review it and add their comments.

### 2.5. Developing the Final Themes

To develop the initial themes for this paper, first S.M. and H.R. met to reorganize the existing codes from all three codebooks. Codes were clustered into areas that had overlapping or similar qualities and characteristics. Once in like groups, the codes were reviewed to determine what content area they covered, discussed with the team to confirm accuracy and relevance, and then were organized into four themes.

*2.6. Trustworthiness Strategies*

Trustworthiness strategies included investigator triangulation in which multiple researchers with diverse backgrounds were involved in the collection and analysis of data. In addition, participants' triangulation was also included by recruiting participants with hip and knee OA and also clinicians working with patients with joint replacement surgery. Participants were asked to provide feedback on a summary of the findings presented by the lead interviewer at the end of each interview/focus group as a form of member checking.

## 3. Results

In this study, 46 participants with hip OA participated in this study (average age = 66.62 [range: 49–85], 32 were women, and 27 had their THA before participating in the study. In addition, 15 family members and one friend (average age = 61.8 [range: 39–82]), 10 were women, also accompanied the participants with hip OA. The second sample consisted of 14 participants with knee OA (average age = 68.17 [range: 55–80]; 8 were women and 10 already had their TKA. The final sample consisted of 15 clinicians (average age = 50.5 [range: 31–67]): 14 were women, 11 physiotherapists, three occupational therapists, and one nurse. On average, they had 17.2 (5-30 years range) years of providing care for patients with hip and knee OA before or after their surgery. Table S2 provides more information about the characteristics of the participants.

Based on the data, four themes emerged: (1) "I didn't get any of that"; (2) "I never got a definitive answer"; (3) "better idea of what's going to happen"; (4) "a lot of people are shifting to online". Table 1 provides an overview of patients' and clinicians' perspectives.

**Table 1.** An overview of patients' and clinicians' perspectives on prehab education.

| | "I Didn't Get Any of That" | "I Never Got a Definitive Answer" | "Better Idea of What's Going to Happen" | "a Lot of People Are Shifting to Online" |
|---|---|---|---|---|
| | **Describes Barriers in Accessing the Prehab** | **Describes Necessary Information in Prehab** | **Describes the Positive and Negative Aspects of Prehab** | **Describes Patient's Perspective on Online Education** |
| **Patients' and caregivers' perspective** | Shortage of facilities, educational sessions, and equipment. | Pain medications and pain management | Informative (positive) | Easy access to information (positive) |
| | Health care provisors lack of time | Exercise and the intensity of the exercises | Receiving information not relevant to their condition (negative) | Doubting the accuracy of the online information (negative) |
| | Embarrass to ask questions | The meaning of recovery | Not receiving enough information (negative) | Reading and seeing negative information (negative) |
| | | Availability of prehabilitation and rehabilitation in their area | | |
| | | Surgery and surgical procedure | | |

**Table 1.** *Cont.*

| | "I Didn't Get Any of That" | "I Never Got a Definitive Answer" | "Better Idea of What's Going to Happen" | "a Lot of People Are Shifting to Online" |
|---|---|---|---|---|
| **Clinicians' perspective** | Patients' limited financial resources | Pain medications and pain management | Better surgery outcome and less anxiety in patients (positive) | Better accessibility (positive) |
| | Patients' mobility problems | Exercise | Lower burden for clinicians working with patients after their Total Hip Replacement (THA) or Total Knee Replacement (TKA) (positive) | Receiving personalized information (positive) |
| | Not a life priority for the patients | Recovery timeline | Sometime patients received too much information (negative) | Less financial burden on health care system (positive) |
| | | Surgery and surgical procedure | | Patients may not be able to ask questions |
| | | Home preparation for after the surgery | | Lack of computer literacy (negative) |
| | | Necessary equipment after the surgery | | Patients may not be able to afford to pay for internet or computer (negative) |

The first column in the table indicates the group.

### 3.1. Theme 1. "I Didn't Get Any of That"
#### 3.1.1. Patients with Hip or Knee OA

Both groups of patients described various reasons for not being able to access the information that they needed regarding their THA or TKA. Family caregivers also reinforced some of the reasons mentioned by patients. The first reason was *shortage of facilities, educational sessions, and equipment*. These shortages were mostly mentioned by patients who lived in rural areas and could not attend educational sessions in other cities because of *long travel distances*:

> *I think it's good, but it can be a little difficult sometimes to arrange. I know for me it's no problem, I live in [Location 2]. But I know that there was a lady that was at the hospital doing the pre-op thing, and she was from [Location 3]. And she was really upset she'd been there all day and she was going to miss her ferry, because the process took her the greater part of the day.* (Man, 67, postoperative, knee OA)

Some patients declared that in the educational sessions they attended, there was a shortage of materials (i.e., books) or equipment and therefore they did not receive the necessary training: *"There was no equipment, there was basically, we were handed the book, the two books I guess, and shown how to wash our hands."* (Man, 66, postoperative, hip OA)

Another problem was health care providers' lack of time, resulting in not being able to provide adequate preoperative training: *"I find I get frustrated in doctor's offices because they seem to be in a hurry, and seem to want to get onto the next client, and therefore information isn't all delivered there."* (Man, 58, preoperative, hip OA). Based on patients' opinions, another substantial barrier was not being referred to the educational sessions:

> *I wasn't referred to anybody. I saw the surgeon, my family doctor referred me to the surgeon, I saw him, I was given the booklet, sent back to [City], and not referred or told about any place that I could access more information.* (Woman, 65, postoperative, hip OA)

*I have to tell him maybe about ten times, you shouldn't do this, and he still is gonna do it the other way. But if I know like from the doctor, and they tell him, "you have to do this to get better", that would help a lot. But every time you go to the doctor, they only have so many minutes to take care for you, and it's very hard for a person who takes care of a sick man.* (Woman, caregiver)

Some even indicated that their healthcare providers were not aware of the THA educational sessions, and they had to initiate the referral themselves.

In addition, few patients in the hip OA group mentioned they were sometimes embarrassed to ask questions, mainly because the questions were personal/private, or they were afraid that the question might sound absurd: *"I think all physicians need to be aware that patients aren't comfortable of sometimes asking. Either through embarrassment issues, or because of a sense that they don't want to appear like an idiot."* (Woman, 59, preoperative, hip OA). Finally, patients said that even when they received the information, it was likely that they forgot the information because there was no reminder or because they received so much information that they could have missed a key part.

### 3.1.2. Clinicians

Similar to patients, clinicians pointed out that patients face various barriers in accessing prehab. The main barrier mentioned by clinicians was patients' limited financial resources that prevented them from either accessing the prehab or following prehab recommendations like buying the necessary equipment. Furthermore, mobility problems were another barrier preventing patients from accessing prehab. Finally, some clinicians believed that receiving prehab is a not a life priority for the patients, especially when patients are marginalized or are low-income.

*They're also the most at-risk of requiring a joint replacement of some kind and they're the ones that are falling through the cracks when receiving the services that they need. They're also the least probably least motivated to do anything about it because they have bigger issues at hand.* (Woman, physical therapist)

In addition to the barriers faced by patients, clinicians also mentioned some challenges they and health care authorities face in providing the prehab for the patients. The main challenges were *financial limitations* and *limited staff time* which decreased the clinician's flexibility in providing prehab, having one-on-one meetings with them, and answering their questions.

*So, it was done live where we connected with [City]. There was a nurse down there that would do half our talk about some of the complications following surgery . . . . that funding was sort of cut. So now, there's actually a video tape about the nurse doing the talk. One of the last talks she did so, or not video tape. It's a DVD. So, I plug that in and the folks, they can watch that for a half hour.* (Woman, occupational therapist)

To solve some of the barriers faced by patients, most clinicians indicated that the education should be provided via in-person one-on-one, or in-person educational group sessions and the information should be provided by clinicians themselves. The main reasons for having in-person educational sessions were so patients could ask questions. This was especially the case for one-on-one educational sessions. However, due to the lack of budget, in most places, patients could not have access to these kinds of sessions. Furthermore, most clinicians believe that the information should be tailored based on the patients' condition and need to receive personalized information.

*I mean, the problem is, there's some, some, groups of people that might need more specific education, and some that, don't like that if you have diabetes or if you're really overweight, you may, you may almost need to have some extra education on that component.* (Woman, nurse)

### 3.1.3. Comparison between Patients and Clinicians

Both groups of patients as well as clinicians were concerned about the distances that patients have to travel to receive prehab and financial limitations that the health care system faces in providing prehab for patients. However, clinicians mentioned prehab may not be a life priority for patients and patients may have financial limitations that disallow them to invest in prehab, two points not mentioned by patients.

*3.2. Theme 2. "I Never Got a Definitive Answer"*

### 3.2.1. Patients with Hip or Knee OA

Data showed that both groups of patients as well as family caregivers struggled to find accurate responses to some of their questions, as one patient explained: *"when I asked, "how does this happen? How come it deteriorated so fast?" I never got a definitive answer."* (Woman, 65, postoperative, hip OA) or as one caregiver explained *"Right, yes, so, yes coming out of the hospital into the car, into the house and what we can let her do and not do".* (Woman, caregiver) They thought they could have received more information on some topics, for example, *"I don't remember specifically receiving much information about pain management, except maybe there was a paragraph or something in one of the information booklets or something."* (Woman, 64, postoperative, knee OA).

Specifically, patients requested information regarding *exercise and the intensity of the exercises* they have to perform before and after surgery: *"It's one thing to look at a picture and go I've got to open my legs, it doesn't say exactly how many times, and how many reps."* (Woman, 59, preoperative, hip OA). Patients also indicated that they needed more information regarding nutrition and weight management: *"I thought maybe a nutritionist actually coming in and giving a little speech might have helped."* (Man, 66, postoperative, hip OA).

*Pain medications and pain management* was another issue brought up by the patients. Patients were concerned with the side effects of pain medications and expressed their desire for receiving more information, as one participant explained:

> *I think one item that perhaps could be included in the pain management discussion would be that if some of the pain medication is in any way addictive to be aware of it and how to perhaps deal with that type of a situation.* (Man, 73, postoperative, hip OA)

Patients also discussed the range of individualized needs that patients will have, and how it is not a one size fits all with pain management:

> *Some people have a low tolerance for pain, other people have a much bigger tolerance for pain. So, I think they've got to adjust it for each, you can't just generically say "I'm going to give this dosage to everybody.* (Man, 67, postoperative, knee OA)

For many patients *the meaning of recovery* was not clear, and they were not certain about the activities that they could perform after their surgery, e.g., *"What should I be making myself do? Day 5 when I get home from the hospital, like should I be walking 100 m?"* (Man, 72, preoperative, hip OA). Patients also indicated needing to know what behaviors and movements they should avoid both before and after surgery and why they should follow such precautions: *"You're not supposed to bend your hip up beyond 90 degrees. But no reason as to why, I want the why."* (Woman, 61, postoperative, hip OA).

Other information that patients were seeking was related to the *availability of prehabilitation and rehabilitation* programs in their communities: *"I had no idea that I could go to pool classes and things in advance of my surgery."* (Man, 60, postoperative, hip OA). Another top issue for patients was related to managing day-to-day activities. The main activities raised included *dressing, transportation, driving, adjusting to weather and safety issues in winter, taking care of their pets after surgery, and facing emergency situations.* Moreover, patients clearly indicated that they needed to gain better personal skills to *manage their psychological wellbeing:* *"There's gotta be more education about calming, calming these fears."* (Man, 66, postoperative, hip OA) and to have realistic expectations: *"their frustration level, it has to all be tempered with the expectation that this isn't going to make you superman or superwomen, it's to alleviate the situation you're in now."* (Man, 67, postoperative, knee OA).

One of the other important topics that patients would like to receive more information about was related to the surgery and surgical procedure. Specifically, about the possible *symptoms and complications* as one patient expressed: *"That my legs would be the same length when I come out?"* (Woman, 49, preoperative, hip OA). Furthermore, participants expressed interest in knowing more about the role of their family caregivers in providing support and what caregivers need to know. They also indicated having information about family caregivers' wellbeing as essential: *"My 18-year-old who's just started college is now the one who's gonna be responsible, and she's already stressing."* (Woman, 55, preoperative, hip OA). Finally, patients, mostly preoperative ones, indicated that they would also like to receive more information about OA and its symptoms: *"I found that there was so many things lacking. Firstly, it was really difficult, the diagnosis, for the osteoarthritis because my G.Ps were not sure where to send me to"* (Woman, 58, postoperative, knee OA).

### 3.2.2. Clinicians

Almost two-thirds of the clinicians emphasized the substantial role of including exercise practices as part of a good prehab. The main reasons for having the exercises as part of the prehab was that they believed knowing about the exercises can help the patients to become more active and have better outcomes after surgery.

> *"I also think it helps them to start the exercises before the surgery so they can, maybe get a little bit stronger, and a little bit more range of motion before they go in for the surgery and have a better outcome after if they're stronger and more mobile."* (Woman, physical therapist)

Some other topics believed to be important to include in a prehab were information related to the necessary equipment after the surgery, managing pain after the surgery, the surgery itself, home preparation for after the surgery, and recovery timeline. The main reasons for providing the information related to these topics was to prepare patients for the surgery and expedite the recovery. In addition, the clinicians also indicated that knowing this information, especially related to the pain management and recovery timeline, can modify patients' expectations. Therefore, in general, clinicians suggested that the above-mentioned topics could prepare the patients physically and emotionally for the journey both before and after surgery: *"I would like to see the surgeon do a detailed uh, discussion of what they do in a surgery, uh so patients have a, a better idea of what's going to happen inside."* (Woman, physical therapist).

### 3.2.3. Comparison between Patients and Clinicians

Both groups of patients as well as clinicians pointed out mostly similar topics that need to be included in prehab. However, clinicians place more emphasis on exercises before surgery while patients would also like to know more about the meaning of recovery, how to overcome their anxiety and fears, and possible symptoms and complications.

### 3.3. Theme 3: "Better Idea of What's Going to Happen"

### 3.3.1. Patients with Hip or Knee OA

Most patients in this study found the prehab *informative*: *"loved them, I thought they were very good. Um, so I think I know enough."* (Man, 72, preoperative, hip OA). However, they mentioned some problems with the education, including hearing information that was *not relevant* to their needs: *"So you can sort of uh, maybe hear a whole lot of things that aren't relevant to you."* (Woman, 76, preoperative, hip OA). Some others indicated that they received contradictory information which confused them: *"There was conflicting information from the people* (healthcare providers) *who have had the preadmission clinic interview."* (Woman, preoperative, hip OA) and finally, some suggested that the prehab they received was inadequate and still they felt *"lost"*.

### 3.3.2. Clinicians

Almost all clinicians had a positive view on prehab and considered prehab as a vital and beneficial resource. Clinicians indicated that prehab is a valuable resource that anyone undergoing THA and TKA should have access to. They also indicated seeing a substantial difference between patients who received the education compared to the ones who had not received any prehab, for example, a clinician (Woman) indicated that "*It is more beneficial. Yeah. I think in general it is good to offer it for anyone right? Because we cannot assume that everyone you know has the right information or gets it.*"

One of the benefits or advantages of prehab for patients mentioned by almost all clinicians in this study was preparing patients for the surgery and for after the surgery. Clinicians indicated that patients who receive prehab know what to expect before and after their surgery and know what they need to do to be better prepared (both emotionally and physically) for their surgery. Clinicians continued to explain that prehab could adjust patients' unrealistic expectations regarding their surgery outcomes and recovery process.

> *I'm sort of saying, they're pretty much different physically. You could do a study and people might say it was a far more positive experience because I knew what I was getting into. So, mentally emotionally they have some definite benefits.* (Woman, occupational therapist)

Another important benefit of prehab reported by clinicians was decreasing patients' anxiety before and after their surgery. The main reason for the reported reduction in anxiety was the patient knew what to expect before and after the surgery and they understood the surgery and the surgical procedure better. Furthermore, clinicians pointed out that the patients who received the prehab are better prepared for their surgery and therefore, are less anxious.

> *It helps decrease any anxiety they have over the surgery. It provides them with the information they may have to prepare. Things to prepare for not only before the surgery but after the surgery to mitigate any concerns, and they know exactly what to anticipate, how the recovery will go, how to take care of themselves, what types of equipment they need, and what to expect from the team at the hospital after the surgery.* (Woman, occupational therapist)

Interestingly, clinicians believed that prehab is beneficial for any clinicians involved in patient care before and after total hip or knee replacement surgeries, including occupational therapists, physical therapists, nurses, and surgeons. They suggested that prehab decreases the burden on clinicians because patients who receive the education are more prepared for their surgery, have information about the care and precautions for after the surgery, and ask less questions about the surgery.

> *So, I think it would just reduce the, the visits and improve the efficiency of the visit, and improve efficiency of, of their care, cause they have more kind of, I guess control or, sense of awareness of, if is it normal or not.".* (Woman, physical therapist)

Finally, echoing patients' concerns regarding prehab, clinicians indicated that for a good prehab, it is unnecessary to overload the patients with too much information as they will not benefit from that and providing too much information may confuse the patients, especially when the information is not relevant to the patients' condition. Moreover, clinicians mentioned that patients do not need to see the actual videos of the surgery, as this might increase anxiety before surgery: "*It may be scary as well provide them information that may not be necessary or again, pertain to their own individual circumstances, so those are some negatives.*" (Woman, occupational therapist).

### 3.3.3. Comparison between Patients and Clinicians

Similar to clinicians, patients pointed out that prehab was in general beneficial for them. Besides the benefits of prehab for patients, clinicians indicate that prehab can be beneficial for other clinicians such as nurses and surgeons. Both groups of patients

and clinicians indicated that prehab in some cases may not be relevant to patients, or they receive too much information. In contrast to patients, however, clinicians did not acknowledge that patients may receive inadequate or contradictory information from various sources.

*3.4. Theme 4: "A Lot of People Are Shifting to Online"*

3.4.1. Patients with Hip and Knee OA

Finally, going online and searching the internet for information was a common approach suggested by patients, e.g., *"I'd seen a couple of things online about things to do for your for you know, if your mobility was decreased."* (Woman, 65, postoperative, hip OA). Participants mentioned that they used the internet to read, watch videos, and listen to online presentations related to OA and hip and knee replacement surgery. One of the main benefits of going online for information was ease of access e.g., *"immediacy, you know instant information vs waiting for an appointment or waiting for a return phone call"* (Man, 60, postoperative, knee OA). While many voiced positive experiences, using online information is not without challenges. One of these challenges was *accuracy of the information*. To express this concern, one patient commented: *"you want reliable sites, and you're hoping that it applies to Canadians as well, but certainly knowing that some procedures can be different."* (Man, 70, preoperative, hip OA). Another problem mentioned by patients was that the internet is encumbered with negative and *gruesome information* e.g., *"The downside is sometimes you get stuff that's really scary* (Man, 70, preoperative) or *"just online stuff, um I didn't want to get into the gory details or visuals or YouTube stuff, I know it's all there about the procedure but it didn't really put me in a better place"* (Man, 60, postoperative, knee OA).

3.4.2. Clinicians

Clinicians responded to the questions related to online education by taking a more academic and generally positive perspective about online education, believing that patients can benefit from it. One of the main benefits of online education was that it provides better accessibility because it is always available, and patients can access it regardless of the time and their geographical location (in contrast to in-person education): e.g., *"I would watch online material that I can access at any time. Because a lot of people end up losing the booklet anyways."* (Man, physical therapist).

Clinicians also indicated that patients can access a wealth of information online based on their needs, a characteristic that cannot be found in in-person education or educational booklets: e.g., *"I think that benefits, that is because, a lot of our, a lot of people are shifting to online resources, so, there are lot of things available online."* (Woman, physical therapist).

Clinicians also indicated online education needs fewer financial resources to maintain and applying this approach can remove the burden from clinicians and staff who otherwise have to provide the in-person education or address their questions, e.g., *"It is that you're not holding up a staff member and we're all just so incredibly busy. So, if you could have a majority of your folks listen to programming online, um that, they're still getting some form of education so, and just out of necessity."* (Woman, occupational therapist).

In addition, clinicians believed most prehab topics can be delivered online, but they also indicated various limitations of online education. The limitations regarding online education were mostly related to its accessibility and not to its content. The most frequently raised issue with online education was patients using online education being unable to ask their questions and unable to receive personalized information. However, clinicians suggested that providing the opportunity for patients to submit their questions can solve this issue to some extent.

> *A benefit is that it's always there and a patient can go back to it whenever they want. And it can include videos of exercises and that kind of thing. And uh, a drawback would be the fact that they can't ask questions.* (Woman, physical therapist)

Computer literacy and having access to computers and the internet were other frequently indicated limitations of online education. Clinicians were concerned that most

patients undergoing THA and TKA are older adults. These patients may have difficulty using a computer or laptop on their own, and even if they could, the other issues of not having computers and the internet exist. They mentioned that most of these patients were retired and cannot afford to buy a laptop, tablet, or other smart device and more importantly cannot afford internet fees, e.g., *"And I guess a challenge is that I see a lot of elderly people who don't actually have a computer or the internet, so it wouldn't work for them."* (Woman, physical therapist).

### 3.4.3. Comparison between Patients and Clinicians

Patients' responses were focused more on informal/non-academic format of online education (e.g., weblogs, conversations on Reddit), including searching information online, reading available information and watching videos online and reading blogs, while clinicians' focus was on formal/academic online education. However, patients and clinicians considered online education as accessible. Considering informal online education, patients had a more negative view of online education, including inaccuracy of the information. However, clinicians were not concerned about the content of a formal/academic online education; their main concern was lack of access to computer and computer literacy.

## 4. Discussion

This qualitative study provided an overview of the patients and clinicians' perspectives of the gaps that exist in current prehab and the barriers that prevent patients from receiving prehab before elective THA/TKA. Moreover, it highlighted both patients' and clinicians' perspectives regarding the current state of prehab.

Patients identified several main reasons for facing difficulties in accessing prehab, including limited access to medical facilities, long travel distances, and clinicians' lack of time. Similar to patients, clinicians also indicated that providing prehab education was not without challenges. The most mentioned challenges by clinicians were patients' financial and mobility issues as well as staff time and health care system's financial limitations. In general, it seems that some of these barriers are universal [27]. For example, clinicians' lack of time is a problem that most patients encounter and for which there is no short-term solution. Other barriers are potential barriers; that is, only some patients may face them. For example, language barriers could be an issue mainly for immigrants. Understanding the universal and potential barriers that patients face in accessing health care is among the very first steps in providing equal access to health care, which is a fundamental human right [28].

Moreover, the findings related to the financial limitations of the health care system in providing prehab, mentioned by both patients and clinicians, are considerable as one of the main goals of prehab education is to reduce the health care cost [29]. As many studies support prehab's role in reducing hospital stays, medical visits and patients' anxiety [10,30], it is important to realize that health care system's financial limitations reduce patients' accessibility to one of the most effective tools minimizing health care cost. While health care authorities aim to use their limited resources on more urgent or severe conditions (e.g., cancer treatment), the question is, is the short-term reduction in expenses associated with prehab education worth the long-term increase in expenses (e.g., more medical visits, complications after surgery, and revision surgery)? Besides the financial limitations faced by the health care system, clinicians indicated that patients might have their own financial limitations that prevent them from investing in prehab, the point that was not highlighted by patients. Considering that for majority of patients the cost of surgery and medical appointments is covered through health insurance in Canada, expenses associated with access to prehab (e.g., travel expenses) may not be the main concern for patients in Canada. Finally, mobility issues were another challenge for patients with OA, as it is well proved that OA is a leading cause of mobility issues in older adults [31]. Therefore, it is understandable that patients with OA, especially the ones who are on a waiting list for THA and TKA, are

facing mobility issues that limit their ability to travel to the prehab educational sessions without any assistance.

While most patients in our study used words such as "fantastic" or "great" to describe their experiences regarding the quality of prehab, they also expressed that for some topics, particularly pain management and recovery, there were gaps. There are several factors that can contribute to patient-perceived educational gaps. First, likely, prehab that patients received did not adequately address a particular topic. For example, caregivers' roles and their well-being may often be ignored in prehab. Second, patients might receive general and not personalized information [32]. While it is important to cover general topics for all patients, patients are more motivated to learn about topics they are personally concerned about [32]. Therefore, it is essential to take patients' perspectives and priorities into account and give them the option to receive personalized education based on their needs.

The topics mentioned by clinicians in order to improve the quality of the current prehab are aligned with the topics mentioned by patients and also the topics currently offered by high-quality prehab education (e.g., OASIS education in Vancouver). However, it is important to mention that most prehab education only focuses on a few topics, usually exercise and precautions [33–35], which are also the preferred topics of clinicians in the current study. Therefore, there is a huge gap between what patients and clinicians consider as important and what is being provided in prehab education. One of the main reasons for the existence of this gap is the limited financial resources available for prehab education, forcing health care authorities to make the decision of how much staff time and resources they can spend on prehab education. In addition, due to lack of resources, clinicians provide prehab to various patient groups (e.g., patients with hip OA and patients with knee OA) simultaneously. Therefore, many patients will receive information that is irrelevant to them. The next question to ask is, how can we decrease the prehab budget but still make it comprehensive, affordable, relevant, and accessible?

The answer to that question is reflected in patients' and clinicians' responses. Patients in our study mentioned using the internet as a way to access prehab, which is consistent with previous studies [36,37]. However, patients also described concerns about the reliability of the information they found on the internet. In a study by Rao and colleagues [36], most patients indicated their preference of getting health information from reliable sources, i.e., their healthcare providers. However, searching the internet for reliable information, understanding the presented information, and evaluating the reliability can be impacted by patients' health literacy [38], so patients with lower levels of health literacy may experience fewer benefits and have difficulty evaluating the accuracy of these resources [39].

Similar to patients, clinicians also pointed out that online health education provides the opportunity for patients to receive a comprehensive, flexible, easily accessible, and cost-effective education [40]. However, we could only find one study in which clinicians' perspective (i.e., nurses) on online prehab education for patients before their THA and TKA were briefly investigated [41]. A website related to prehab education for THA and TKA patients was created by the authors, and the results of this study showed that nurses had a positive perspective towards the website. The nurses indicated the accessibility of the website would have a positive impact on the patients and also having patients review the website before their prehab education sessions with the nurse can increase the nurse's efficiency because they would only need to reinforce the prehab education briefly.

While online prehab education sounds like a reasonable solution to increase accessibility and decrease expenses, it should be indicated that it is not without drawbacks. As indicated by patients and clinicians, patient's computer literacy and having access to the internet and computer can be a hindrance for online education. However, based on the 2016 data from Statistics Canada [42], internet use was 84.95%, 75.30%, and 62.02% for Canadians aged between 65–69, 70–74, and 75–79, respectively. Therefore, while not all senior Canadians use the internet, on average more than 70% of them have access to the internet and can benefit from online education.

The current study had several limitations that need attention. We made an effort to recruit patients and clinicians from both urban and rural areas; however, the majority of our participants lived in urban areas, which may impact the transferability of our findings. Furthermore, the majority of our participants were females, while the Canadian data show that 55% of patients undergoing THA are females [8], again limiting the transferability of findings. Likely, males' perspectives and educational needs were not sufficiently captured. Number of patients with hip OA was higher than the patients with knee OA and the clinicians. However, the patients with hip OA participated in focus groups while patients with knee OA and clinicians participated in the semi-structured interviews which provided them the opportunity to express their perspective individually. We only recruited family caregivers of patients with hip OA and therefore we do not have information on the perspective of family caregivers of patients with knee OA. Moreover, we did not record information related to the outcome of the potential complications of the surgery. Therefore, we are not able to evaluate whether participants' experiences were related to their surgery outcome. Finally, majority of the clinicians were physiotherapists which should be considered when interpreting the finding of this study.

## 5. Conclusions

The current study manifests that patients and clinicians agree that prehab should be comprehensive and encompass a wide variety of topics from before surgery to after surgery. Finally, patients and clinicians indicated that an online platform is an appropriate vessel for providing comprehensive, accessible, and affordable prehab education to patients.

**Supplementary Materials:** The following are available online at https://www.mdpi.com/article/10.3390/disabilities1040025/s1, Table S1: Interview guide used in the mixed focus group, Table S2: The demographic information of patients with hip and knee OA.

**Author Contributions:** Conceptualization, S.M., W.C.M., W.W., M.C., J.M.R. and M.W.; methodology, S.M., W.C.M. and, J.M.R.; formal analysis, S.M., H.R., W.C.M., W.W., M.C., J.M.R. and M.W.; writing—original draft preparation, S.M.; writing—review and editing, S.M., H.R., W.C.M., W.W., M.C., J.M.R. and M.W.; supervision, S.M., W.C.M. and J.M.R.; project administration, W.C.M.; funding acquisition, W.C.M. All authors have read and agreed to the published version of the manuscript.

**Funding:** This project was funded by a Vancouver Coastal Health Research Institute (VCHRI) team grant (PI: W.W.). S.M.'s contribution was supported by a CIHR Project Grant (#364553; PI: W.C.M.).

**Institutional Review Board Statement:** The study was conducted according to the guidelines of the Declaration of Helsinki and approved by the Institutional Review Board of the University of British Columbia (knee OA study sample (KNEEDS): H18-01417; Clinician Sample: H18-01343; Hip OA sample (HHIP): H15-03410)).

**Informed Consent Statement:** Informed consent was obtained from all subjects involved in the study.

**Data Availability Statement:** The data presented in this study are available on request from the corresponding author. The data are not publicly available due to the qualitative nature of the study and the restricted permission from the participants.

**Conflicts of Interest:** The authors declare no conflict of interest.

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
