# Peer review of "A Qualitative Study on Prehabilitation before Total Hip and Knee Arthroplasties: Integration of Patients’ and Clinicians’ Perspectives"

_disabilities, doi:10.3390/disabilities1040025_

Round 1
Reviewer 1 Report
This is a very interesting study providing an overview of patients and caregivers perception over the prehabilitation programs for patients with hip or knee osteoarthritis that are scheduled or already underwent surgery. The study intention is to identify the gaps in the existing prehabilitation programs. The results confirm that there is a need for prehab programs and that the use of online means might overpass some of the drawbacks of current prehab options.
I have some comments and following suggestions:
Line 92 - Eligibility Criteria: The eligibility criteria does not include any reference to the surgery itself and postoperative outcome. It would be necessary to mention if some ore none of the included subjects suffered from any complications during or after surgery. This might be influential over their global perspective.
Does the time window (before or after surgery) has any influence over the patient’s perception of the entire event?
Line 105: Why did the researchers chose to use focus groups for THA and interview for TKA and clinicians? Why not interview for all? In focus groups the answer provided by one subject may be influential over the others. The authors shall provide an explanation for their choice of method.
Line 161: therefore QD, repeats in the same sentence
Line 168: NVIVIO should be defined properly as: NVivo qualitative data analysis Software (only the first two letters in capitals). In this format it may be confused with an abbreviation. Also, it is custom to mention the version of the software.
Line 275: Is it, “the nurse doing the walk” or “the nurse doing the talk”?
Line 545: The statement: “most patients in our study were satisfied” should be first stated in the results section (if applicable) and only afterwards discussed in the Discussions section
Author Response
Dear Reviewer,
We sincerely appreciate the comments and feedback that we received from you. Below, we explained how we addressed each comment:
This is a very interesting study providing an overview of patients and caregivers perception over the prehabilitation programs for patients with hip or knee osteoarthritis that are scheduled or already underwent surgery. The study intention is to identify the gaps in the existing prehabilitation programs. The results confirm that there is a need for prehab programs and that the use of online means might overpass some of the drawbacks of current prehab options.
I have some comments and following suggestions:
- Line 92 - Eligibility Criteria: The eligibility criteria does not include any reference to the surgery itself and postoperative outcome. It would be necessary to mention if some ore none of the included subjects suffered from any complications during or after surgery. This might be influential over their global perspective.
Response: This is an interesting comment. However, we did not record information related to the surgery outcome. We added this as one of the limitations of our study.
Does the time window (before or after surgery) has any influence over the patient’s perception of the entire event?
Response: Because of the qualitative nature of this study, we did not evaluate the impact of surgery timeline on patients’ perceptions. However, we noticed that post-op patients would point out to the positive aspects of pre-habilitation what was helpful for them, and the questions that they would like to receive the answer for while pre-op patients mentioned the questions that they still had at the time of the interview. However, this was not a systematic observation. For that reason, we did not present this information in this manuscript.
- Line 105: Why did the researchers chose to use focus groups for THA and interview for TKA and clinicians? Why not interview for all? In focus groups the answer provided by one subject may be influential over the others. The authors shall provide an explanation for their choice of method.
Response: We started this line of research by recruiting THA patients. After conducting the THA project, we added two new projects that is TKA and clinicians. TKA and clinicians’ projects were smaller scope projects and to improve the efficiency and feasibility of the recruitment, we conducted semi-structured interviews. This method provided a better flexibility, especially for the clinicians to participate in this study. We added this information on page 3, lines 132-135.
- Line 161: therefore QD, repeats in the same sentence.
Response: Thank you, we removed the additional QD.
- Line 168: NVIVIO should be defined properly as: NVivo qualitative data analysis Software (only the first two letters in capitals). In this format it may be confused with an abbreviation. Also, it is custom to mention the version of the software.
Response: Thank you. We addressed this issue.
- Line 275: Is it, “the nurse doing the walk” or “the nurse doing the talk”?
Response: We corrected the misspelling.
- Line 545: The statement: “most patients in our study were satisfied” should be first stated in the results section (if applicable) and only afterwards discussed in the Discussions section
Response: We removed “most patients in our study were satisfied” and modified the sentence accordingly.
Reviewer 2 Report
This is an interesting study investigating the perspectives of patients with hip or knee AO but also of clinicians. Please include the clinicians in the purpose of the study.
Furthermore, I would include a table of the patients' demographics and a copy of the questionnaire. This does not need to be part of the manuscript but should be submitted as part of the supplemental data.
Line 133, 134 since Disabilities is an unblinded journal please delete the authors 1,-6 and write the abbreviations.
It is interesting to show what the individuals pointed out, I would suggest to include an algorithm figure in how to change the concept.
Author Response
Dear Reviewer,
We sincerely appreciate the comments and feedback that we received from you. Below, we explained how we addressed each comment:
- This is an interesting study investigating the perspectives of patients with hip or knee AO but also of clinicians. Please include the clinicians in the purpose of the study.
Response: Thank you for your comment. We modified the purpose section based on this comment.
- Furthermore, I would include a table of the patients' demographics and a copy of the questionnaire. This does not need to be part of the manuscript but should be submitted as part of the supplemental data.
Response: We added the requested information as appendices.
- Line 133, 134 since Disabilities is an unblinded journal please delete the authors 1,-6 and write the abbreviations
Response: Thank you for your comments. We added the initials to the text.
Reviewer 3 Report
Review Comments for the Author(s)
Journal: Disabilities
Manuscript Number: disabilities-1364044
Title: A qualitative study on prehabilitation before total joint arthroplasties: Integration of patients’ and clinicians’ perspectives
Overview: This is a study reflecting the perspectives of patients with hip and knee OA, their caregivers, and clinicians who are working with these patients about current prehab educational content and delivery.
Comments:
- Title
The title is okay, but might be clearer if you add knee and hip, for example:
A qualitative study of patients’ and clinicians’ perspectives on prehabilitation before total knee and hip arthroplasties.
- Abstract
In the abstract on line 23, you wrote prehab, please wrote preoperative rehabilitation (“prehab”) the first time you use the shortening. Try to be consequent were you put the parentheses, on line 24 before OA.
- Keywords
Please add total knee replacement.
- Introduction and aim
The introduction is a little bit messy and needs to be extended.
My suggestion is to start with the definition of prehabilitation, including the timeframe (does it start when the referral is sent to the orthopedic clinic, then the patients are put on the waiting list for surgery, or when they get a surgery appointment).
Are there different definitions of prehabilitation (exercise, education, quit smoking, quit drinking alcoholic, lose weight, etc)?
What is the main aim of prehabilitation? Anxiety before and after surgery, get a better outcome, fewer complications, etc? Is there evidence for fewer complications in patients that quit smoking, drink alcoholic, lose weight, physical activity for example before surgery with total knee/hip OA? Try to describe how important it is with prehabilitation.
Should patients with knee and hip OA get the same prehabilitaion?
Who needs prehabilitation, patients, family caregivers? Why?
Who educates in prehabilitation? Doctors, nurses, physios, occupational therapists, patients? If you have decided to explore a special prehabilitaion delivered of a physio describe that.
Describe the different ways to deliver prehabilitation, face-to-face, via phone, paper, internet with adequate references if there are references otherwise tell that the evidence is lacking.
End the introduction with all the gaps and argue why you want to do this study and that this study can contribute with.
I think most of the information already is in the introduction but needs to be clearer, so it is easy for the reader to understand why this is an important study.
- Materials and methods
The material included three samples:
- patients with hip OA and their family caregivers
- patients with knee OA
- clinicians who were working with patients with hip and knee OA
Please describe the reason for the three groups, why should patients with hip OA have their family caregivers with them but not patients with knee OA?
In the text you describe to reduce the potential expenses, family caregivers of patients with knee OA were not recruited. I believe that patients with knee and hip replacement are quite different and need different stuff, this needs to be addressed in the discussion.
On line 127 you describe: “Each interview (45-60 minutes) and focus group (90 minutes) consisted of two main parts: the first, being preferred content and the learning preferences of patients with hip or knee OA in respect to presurgical education, and the second, focusing around the patients’ preferred format for the presurgical education (e.g. images, text, videos) and feedback on an eHealth tool used to deliver prehab (more details about the second part have been provided in [23].”
I wonder if the patients were given different kinds of formats images, text, videos, eHealth), before they were discission them? Please describe.
Results
In the result part you describe the 3 groups:
- Patients with hip OA
- Patients with knee OA
- Clinicians
Please be consequent how you describe the patients in groups 1 and 2, total (n=xx), woman (n=xx), THA (n=xx), participate in prehabilitaion, etc, that make it much easier to read.
On line 210, you describe 15 clinicians, describe which profession they belong to (physiotherapists, nurses, occupational therapists etc)
Discussion
On line 509, “This qualitative study provided an overview of the gaps that exist in current prehab and the barriers that prevent patients from receiving prehab before elective THA/TKA”. Is this true? Is this an overview of the gaps or the patient's and clinician's thoughts about the gaps?
In the limitation part, you need to mention that you have more patients with hip OA than knee OA, did that affect the results? You did only have family caregivers for patients with hip OA.
Were all professions represented by the 15 clinicians or did you include more physiotherapists than nurses etc? I guess that will affect the result, please describe.
Conclusion
The conclusion is very long, please consider what you want to tell the reader, what are the most important findings in this stud? Do the conclusion easy to read.
Author Response
Dear Reviewer,
We sincerely appreciate the comments and feedback that we received from you. Below, we explained how we addressed each comment:
Review Comments for the Author(s)
Journal: Disabilities
Manuscript Number: disabilities-1364044
Title: A qualitative study on prehabilitation before total joint arthroplasties: Integration of patients’ and clinicians’ perspectives
Overview: This is a study reflecting the perspectives of patients with hip and knee OA, their caregivers, and clinicians who are working with these patients about current prehab educational content and delivery.
Comments:
- Title
- The title is okay, but might be clearer if you add knee and hip, for example: A qualitative study of patients’ and clinicians’ perspectives on prehabilitation before total knee and hip arthroplasties.
Response: We added “hip and knee” to the title to make it clearer.
- Abstract
- In the abstract on line 23, you wrote prehab, please wrote preoperative rehabilitation (“prehab”) the first time you use the shortening. Try to be consequent were you put the parentheses, on line 24 before OA.
Response: Thank you for pointing out to this issue. We modified the text accordingly.
- Keywords
- Please add total knee replacement.
Response: This point has been addressed.
- Introduction and aim
- The introduction is a little bit messy and needs to be extended.
My suggestion is to start with the definition of prehabilitation, including the timeframe (does it start when the referral is sent to the orthopedic clinic, then the patients are put on the waiting list for surgery, or when they get a surgery appointment).
Are there different definitions of prehabilitation (exercise, education, quit smoking, quit drinking alcoholic, lose weight, etc)?
What is the main aim of prehabilitation? Anxiety before and after surgery, get a better outcome, fewer complications, etc? Is there evidence for fewer complications in patients that quit smoking, drink alcoholic, lose weight, physical activity for example before surgery with total knee/hip OA? Try to describe how important it is with prehabilitation.
Should patients with knee and hip OA get the same prehabilitaion?
Who needs prehabilitation, patients, family caregivers? Why?
Who educates in prehabilitation? Doctors, nurses, physios, occupational therapists, patients? If you have decided to explore a special prehabilitaion delivered of a physio describe that.
Describe the different ways to deliver prehabilitation, face-to-face, via phone, paper, internet with adequate references if there are references otherwise tell that the evidence is lacking.
End the introduction with all the gaps and argue why you want to do this study and that this study can contribute with.
I think most of the information already is in the introduction but needs to be clearer, so it is easy for the reader to understand why this is an important study.
Response: Taking the points mentioned by the reviewer, we modified the introduction.
- Materials and methods
The material included three samples:
- patients with hip OA and their family caregivers
- patients with knee OA
- clinicians who were working with patients with hip and knee OA
- Please describe the reason for the three groups, why should patients with hip OA have their family caregivers with them but not patients with knee OA?
Response: We started this line of research by recruiting THA patients. After conducting the THA project, we added two new projects that is TKA and clinicians. TKA and clinicians’ projects were smaller scope projects and to improve the efficiency and feasibility of the recruitment, we conducted semi-structured interviews. This method provided a better flexibility, especially for the clinicians to participate in this study. We added this information on page 3, lines 132-135.
- In the text you describe to reduce the potential expenses, family caregivers of patients with knee OA were not recruited. I believe that patients with knee and hip replacement are quite different and need different stuff, this needs to be addressed in the discussion.
Response: We addressed this point in our discussion (page 13, lines 578-581) as also as part of our limitation (page 14, lines 620-623).
- On line 127 you describe: “Each interview (45-60 minutes) and focus group (90 minutes) consisted of two main parts: the first, being preferred content and the learning preferences of patients with hip or knee OA in respect to presurgical education, and the second, focusing around the patients’ preferred format for the presurgical education (e.g. images, text, videos) and feedback on an eHealth tool used to deliver prehab (more details about the second part have been provided in [23].”
- I wonder if the patients were given different kinds of formats images, text, videos, eHealth), before they were discission them? Please describe.
Response: We did have different type of videos, different type of questions, different images in the module that participants viewed. We provided more detailed about that section of study in [23]. We added brief information related to the second part to the text, however, the data presented in this manuscript is only based on the first part of the focus groups/interviews (prior to the observation of the mock eHealth module).
Results
In the result part you describe the 3 groups:
- Patients with hip OA
- Patients with knee OA
- Clinicians
- Please be consequent how you describe the patients in groups 1 and 2, total (n=xx), woman (n=xx), THA (n=xx), participate in prehabilitaion, etc, that make it much easier to read. On line 210, you describe 15 clinicians, describe which profession they belong to (physiotherapists, nurses, occupational therapists etc)
Response: We modified the description based on the suggestions.
Discussion
- On line 509, “This qualitative study provided an overview of the gaps that exist in current prehab and the barriers that prevent patients from receiving prehab before elective THA/TKA”. Is this true? Is this an overview of the gaps or the patient's and clinician's thoughts about the gaps?
Response: We modified this sentence to indicate that the study reflects the patients and the clinicians’ perspectives of the gaps.
- In the limitation part, you need to mention that you have more patients with hip OA than knee OA, did that affect the results? You did only have family caregivers for patients with hip OA.
Response: We modified our limitations to reflect for these issues.
- Were all professions represented by the 15 clinicians or did you include more physiotherapists than nurses etc? I guess that will affect the result, please describe.
Response: Majority of the participated clinicians were physiotherapists. We added this information in the results section and also mentioned this as part of our limitations.
Conclusion
- The conclusion is very long, please consider what you want to tell the reader, what are the most important findings in this stud? Do the conclusion easy to read.
Response: Thank you for your comment. We shortened the conclusion and made it easier to read.
Round 2
Reviewer 2 Report
thanks for the adjustments. In my opinion the manuscript can be published in the present form.
Reviewer 3 Report
Dear authors, I do not have any more comments to add.